# Testing geometric representation hypotheses from simulated place cell recordings

**Thibault Niederhauser**                                    TNIEDERH@STUDENT.UBC.CA
*School of Engineering, École Polytechnique Fédérale de Lausanne*
*School of Biomedical Engineering, University of British Columbia*

**Adam Lester**                                    ADAM.LESTER@UBC.CA
*School of Biomedical Engineering, University of British Columbia*

**Nina Miolane**                                    NINAMIOLANE@UCSB.EDU
*Electrical and Computer Engineering, University of California, Santa Barbara*

**Khanh Dao Duc**                                    KDD@MATH.UBC.CA
*Departments of Mathematics and Computer Science, University of British Columbia*

**Manu S. Madhav**                                    MANU.MADHAV@UBC.CA
*School of Biomedical Engineering, Djawad Mowafaghian Centre for Brain Health, University of British Columbia*

**Editors:** Sophia Sanborn, Christian Shewmake, Simone Azeglio, Arianna Di Bernardo, Nina Miolane

## Abstract

Hippocampal place cells can encode spatial locations of an animal in physical or task-relevant spaces. We simulated place cell populations that encoded either Euclidean- or graph-based positions of a rat navigating to goal nodes in a maze with a graph topology, and used manifold learning methods such as UMAP and Autoencoders (AE) to analyze these neural population activities. The structure of the latent spaces learned by the AE reflects their true geometric structure, while PCA fails to do so and UMAP is less robust to noise. Our results support future applications of AE architectures to decipher the geometry of spatial encoding in the brain.

**Keywords:** Place Cells, Task-Specific Encoding, Representation Learning, Autoencoders

## 1. Introduction

Animals including humans navigate by representing spatial locations in the combined neural activity of the hippocampal formation. Place cells represent spatial locations by firing when the animal is at a specific location in physical Euclidean space (at particular $(x, y)$ coordinates) (O'Keefe and Dostrovsky, 1971; O'Keefe and Nadel, 1978). However, recent work has shown that when animals perform complex tasks, these neurons represent 'places' in spaces that are task-relevant but not always Euclidean (Behrens et al., 2018; Whittington et al., 2020). We are interested in developing a methodology that can resolve this apparent divide about the role of place cells. To do so, we explore spatial navigation in the context of graphs, as simple topological objects defined only by the adjacency of a finite number of nodes connected by edges. In a physical maze apparatus, we will present various routes that share a common underlying graph topology to a moving rat while recording from hippocampal place cells. Across trials, the topology of graphs remains the same whereas their physical Euclidean locations change. In this work, we simulate the activity of place cells

whose firing rates conform to one of two competing hypotheses: they represent locations in Euclidean space as classically defined, or they represent locations in graph space, reflecting the structure of the task. Upon using manifold learning methods such as UMAP and Autoencoder networks to encode the simulated neural activity in a latent space, we examined how such a latent representation can be identified with the two hypotheses.

## 2. Methods

### 2.1. Behavioural and neural data generation

Our neurophysiological experiments will utilize a novel dynamic maze apparatus, which consists of a 7×7 grid of octagonal tiles (Figure 1a). Any desired maze configuration can be formed by opening and closing specific gates (Figure S1). In the maze, rats will navigate to goal positions to receive reward. We simulated these goal-seeking behavioural trajectories and corresponding place cell firing data. For a given configuration, we modeled the motion of a rat by setting up a goal position $g \in \mathbb{R}^2$ and simulating a discrete random walk that is biased towards $g$ (detailed in Appendix A). We simulated the time series of the neural state associated with such a trajectory as the firing rate of $N$ neurons. To do so, we first assigned a specific field center position $c$ to each neuron. For a given position $x$ of the rat, the firing rate $f(x, c)$ is

$$f(x,c) = F_{max}\left(\exp\left[-\frac{1}{2}\sqrt{\frac{d(x,c)}{\sigma}}\right] + \varepsilon\mathcal{N}(0,1)\right),$$

where $d$ is a distance function over the maze, $F_{max} > 0$ is the maximum firing rate of the neuron, $\sigma > 0$, $\varepsilon \geq 0$ and $\mathcal{N}(0,1) > 0$ is Gaussian white noise. In practice, we used $F_{max} = 40Hz$, $\sigma = 0.3$ and $\varepsilon = 0$ or $\varepsilon = 0.1$ (for additive noise). $\sigma$ parameterizes the place field size. Parameters were chosen so that the fields have biologically realistic firing rates and areas (Huxter et al., 2003; Neher et al., 2017). The distance function $d(.)$ depends on the geometry that the place fields represent, which is derived from one of the two hypotheses:

1. *Euclidean hypothesis*: Neurons fire according to 2D Euclidean distance from the field center, such that $d = \|\cdot\|_2$, or $d(x,y) = \sqrt{||x-y||^2}$.

2. *Graph hypothesis*: Neurons fire locally according to the Euclidean distances as in 1, but within a range defined by the number of tiles connecting $x$ and $y$. That is, $d(x,y) = \sqrt{||x-y||^2}$ if $d_g(x,y) \leq 1$ and $\infty$ elsewhere, where $d_g(x,y)$ is the number of tiles in the shortest path along the graph between $x$ and $y$ (Figure 1b, second panel).

Under the Euclidean hypothesis, the place field centers $c$ are distributed randomly across the whole platform (Figure 1b) in a particular maze configuration, and remain in the same physical position across configurations. Under the graph hypothesis, the physical positions of the place fields change across configurations, to preserve the same relative position in the graph (see Appendix B).

### 2.2. Autoencoder design

We applied Autoencoders (AEs) (Tschannen et al., 2018) to our datasets. The AEs take a neural state $S \in \mathbb{R}^N$ as input (rescaled firing rate of $N$ neurons $\in [0,1]^N$) and learn an

embedding in a 3-dimensional latent space (sufficient to capture the 2D Euclidean space and any graph topology in our apparatus). The encoder and decoder are fully connected networks, with 4 hidden layers of sizes 64, 32, 16 and 8, and leaky ReLU activation functions. The encoder and decoder output layers are respectively linear and sigmoidal. The data is split into training (70%), validation (20%) and testing (10%) sets, with training performed in batches of 32 data points, using the ADAM optimizer.

### 2.3. Implementation and Code Availability

The code for simulations was developed using Python 3.8.0. The Autoencoder design and training was performed using PyTorch 1.11.0 and PyTorch-Lightning 1.6.5. The code and datasets are available in our Github repository.

## 3. Results

### 3.1. Simulating place field cells firing according to different representations

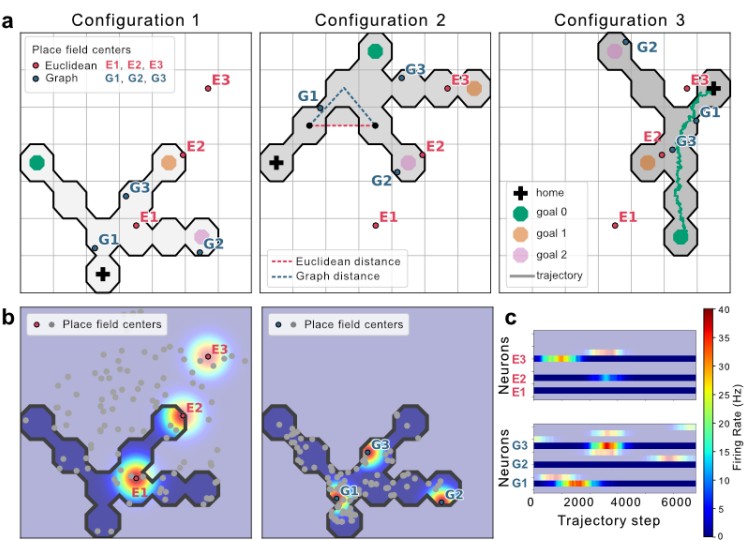

**Figure** 1: Maze configurations and simulated firing. **(a)** Three maze configurations showing place field centers of three example neurons for both hypotheses, Euclidean distance and $d_g(.)$ (here $d_g(.) = 2$), and an example simulated trajectory. **(b)** Place fields of the three neurons under both hypotheses. **(c)** Firing rates of 10 neurons for both hypotheses, for the trajectory shown in **(a)**.

We considered 25 maze configurations (3 examples shown in Figure 1a). While different in shape, these mazes describe the same graph topology, which consists of a home tile connected to a central choice tile that connects to 3 other goal tiles. The goal tiles sometimes overlap across configurations in Euclidean space (e.g. goals 2,3,2 in configurations 1,2,3, resp.), which allows for dissociating if the neural representation conforms to the Euclidean or graph hypothesis. To compare the firing rates produced under these two hypotheses, we

considered two sets of $N = 100$ place cells sampled across the grid. Our final dataset thus consisted of the firing rates of 100 neurons for 2 trajectories $\times$ 3 goal locations $\times$ 25 maze configurations $\times$ 2 hypotheses $\times$ 2 noise conditions ($\varepsilon = 0$ and $\varepsilon = 0.1$) (e.g. Figure 1c).

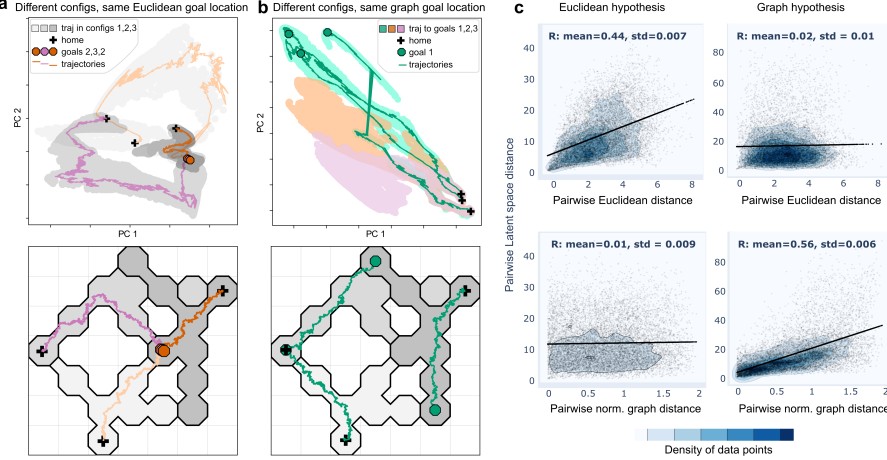

**Figure** 2: Latent space comparison **(a)** AE Latent states learned from the Euclidean dataset. Trajectories converge towards the same goal in the latent space and in the maze configurations. **(b)** Same as **a**, for the Graph dataset. **(c)** Comparison between pairwise latent space distances and pairwise Euclidean and normalized graph distances for 100 sets of $10^4$ pairs of points sampled from each dataset.

### 3.2. Using an Auto-Encoder to represent neural firing rate geometry

We separately trained two AEs, described in Section 2.2, on the graph and Euclidean datasets without noise ($\varepsilon = 0$), to learn a representation of the neural states within a latent space (shown in Figure S2). As we set the dimension of the latent space to 3, we found that after running standard PCA, 96.3% and 95.8% of the variance of the data in the graph and Euclidean AE latent space were explained by the first two principal components (PC's), suggesting that the rat's motion on the 2D platform is well captured. We further confirmed this result by visualizing the latent states associated with the trajectories on the 2 PC's in Figure 2a-b. For the Euclidean data, the goal location shared across all configurations appears as a single location in the latent space (see Figure 2a). Similarly, for the graph data, the same goal, although being located at different positions across configurations, appears at relatively close positions in the latent space (see Figure 2b). Finally, we compared in Figure 2c the pairwise Euclidean distances in the latent space (for 100 sets of $10^4$ sampled pairs of data points) to the corresponding pairwise Euclidean and normalized graph distances in the maze (for a definition of the normalized graph distance, see Appendix C). With a better correlation achieved when using the Euclidean (normalized graph) distance for the Euclidean (graph) dataset, this comparison shows that the standard metric in the learned latent space reflects the choice of the neural representation. In contrast, we failed to observe such a separation using PCA instead of AEs (see Figure S5).

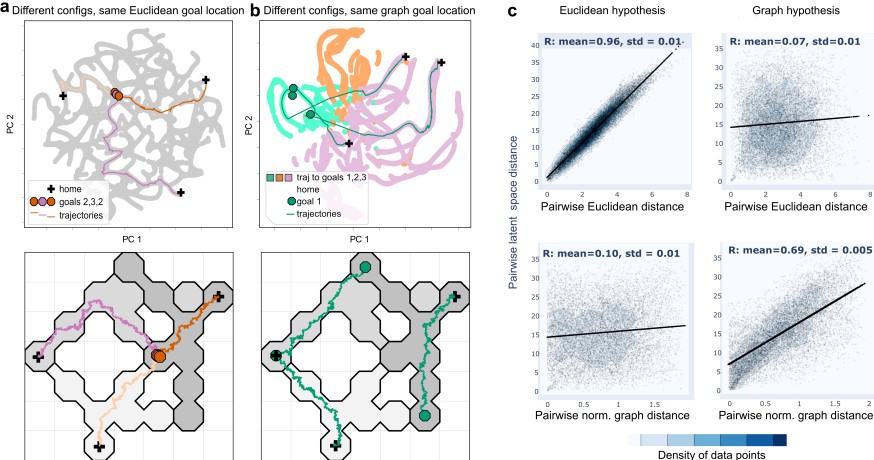

**Figure** 3: UMAP Latent space comparison **(a)** Latent states learned from the Euclidean dataset. Trajectories converge towards the same goal in the latent space and in the maze configurations. **(b)** Same as **a**, for the graph dataset. **(c)** Comparison between pairwise latent space distances and pairwise Euclidean and normalized graph distances.

### 3.3. Using UMAP to represent neural firing rate geometry

We used UMAP (McInnes et al., 2018) to embed neural data into a 2-dimensional latent space. Trajectories and pairwise distances show results comparable to the AEs. Similarities between trajectories towards the same goal are less apparent (Figure 3a-b). Correlations obtained with the pairwise distance metrics are better (Figure 3c). UMAP fails to capture the structure in noisy Euclidean data while it does not degrade severely in noisy graph data (see Figures S6 and S7). The Euclidean data is sparser since place field centers are distributed across the whole platform rather than a subset of tiles. This suggests that AEs are better suited for representing the structure of sparser codes in noisy real-world data.

### 4. Conclusion

We presented two models of place cells in a maze, associated with Euclidean or graph representations. Applying UMAP and AE to a set of 25 maze configurations, we found that the latent representation reflected the geometry inherent to the way population neural activity was simulated. This suggests that UMAP and AE architectures can discriminate between geometric representations and outperform linear methods such as PCA. AEs may outperform UMAP in the presence of noise. We will more thoroughly explore the performance of different architectures (e.g. Miolane and Holmes (2019)) on more complex tasks; this would require a more refined and parametric quantification of the geometry and topology of the latent space, beyond the correlations presented in this work. In addition, a more realistic model of neural activity should be considered, by taking into account remapping of the place cells, path integration, and temporal dynamics.

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

## Appendix A. Simulating rat trajectories using a biased random walk

For a given maze configuration and goal location $g \in \mathbb{R}^2$, we generate a rat moving trajectory as a discrete-time random walk over the maze. More precisely, given the rat at position $x(n)$ ($n \in \mathbb{N}$), we simulate its next position as $x(n+1) = x(n) + l\boldsymbol{v}$, where $l \sim U(l_{min}, l_{max})$ is the displacement length, uniformly sampled between $0 < l_{min} < l_{max}$, and $\boldsymbol{v}$ is a random unit vector, such that

$$\boldsymbol{v} = \begin{cases} \boldsymbol{v_g}(x(n)) & \text{w.p. } p_{drift} \\ \boldsymbol{v_r} & \text{w.p. } 1 - p_{drift}, \end{cases} \tag{1}$$

where $p_{drift} \in (0,1)$ is the probability that the rat moves towards the goal, with $\boldsymbol{v_g}(x(n))$ being the unit vector towards the centre of the next tile to be taken on the shortest path from $x(n)$ to $g$, and $\boldsymbol{v_r}$ being a random unit vector whose direction is uniformly sampled over $(0, 2\pi)$. In practice, we set $p_{drift} = 0.1$, $l_{min} = 0.009$, $l_{max} = 0.011$ and the maze octagonal tiles are inscribed in $1 \times 1$ squares (for some examples of trajectories, see Figures 1a and 2b).

## Appendix B. Sampling place field locations under the graph hypothesis

Under the Euclidean hypothesis, we sample the place cell centers over the whole grid once, so they remain invariant with respect to the maze configurations. However, under the graph hypothesis, each center that gets sampled has to remain invariant according to the graph topology, and thus translates, rotates and scales accordingly across the maze configurations. This section describes our procedure to sample the centers under the graph hypothesis. With all the mazes in our study mapping to the same graph topology (see Section 3.1 and Figure 1a), we first label the coordinates of the tile centers as (see Figure S3):

- $v_0$ the center of the home tile.

- $v_1$, $v_2$ and $v_3$ the centers of the goal tiles.

- $v_4$ the center of the central tile.

We then introduce the 4 "edge" vectors $(\boldsymbol{e_i}) = \overrightarrow{v_4 v_i}$, with the corresponding direct orthogonal unit vectors $(\boldsymbol{e_i^{\perp}})$. For any maze configuration, we sample a place field center as $c = v_4 + \delta \boldsymbol{e_k} + \rho \boldsymbol{e_k^{\perp}}$, where

- $k \sim U(\{0,1,2,3\})$ is the index of the sampled edge $\boldsymbol{e_k}$.

- $\delta \sim U(0,1)$ is the relative distance of the cell along the sampled edge.

- $\rho \sim U(-1,1)$ is the lateral shift associated with the sampled edge.

## Appendix C. Normalized graph distance

We introduce a normalized graph distance to measure the graph distance between two points located in different maze configurations from our study. For two points $x$ and $y$ in different maze configurations, with corresponding edge indices ($k_x$ and $k_y$), and relative positions

along edges ($\delta_x$ and $\delta_y$) (see Appendix B), the normalized graph distance between $x$ and $y$ is given by

$$d_{\text{norm}}(x, y) = \begin{cases} |\delta_x - \delta_y|, & \text{if } k_x = k_y \\ \delta_x + \delta_y, & \text{if } k_x \neq k_y \end{cases} \tag{2}$$

**Supplementary Figures**

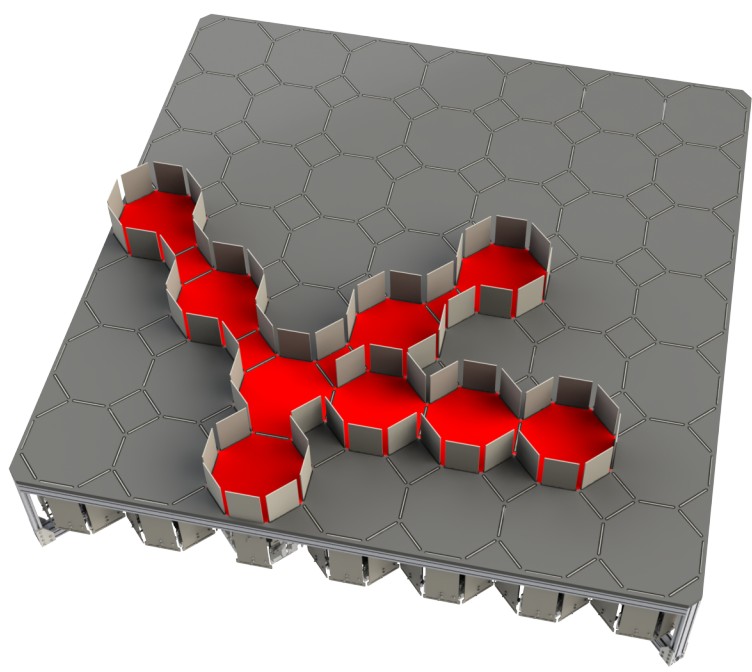

Figure S1: Maze apparatus. The apparatus consists of a 7×7 grid of octagonal tiles. Gates can be opened and closed to form any desired maze configuration. Configuration 1 (Figure 1a) is currently activated and the portion of the maze available to the rat is highlighted in red for illustrative purposes

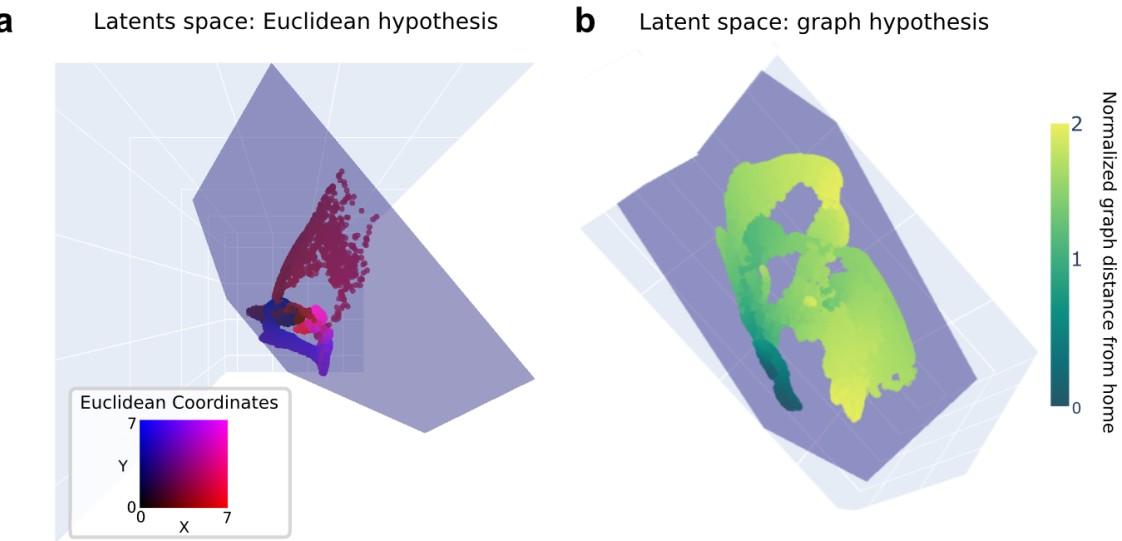

Figure S2: Latent states in $\mathbb{R}^3$. **(a)** Latent space of the Euclidean dataset, color-coded with the Euclidean positions on the maze platform. **(b)** Latent space of the graph dataset, color-coded with the normalized graph distance from home.

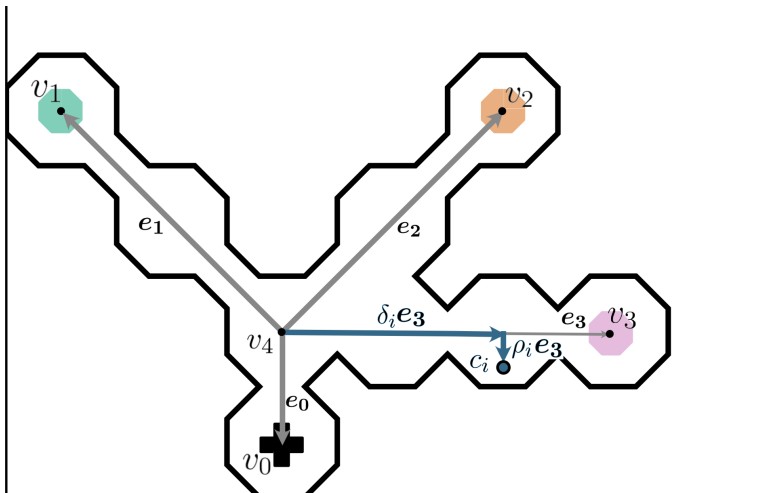

Figure S3: Example of a place field center $c_i$ in maze configuration 1. The edge index is $k_i=3$, the relative distance on the edge is $\delta_i = 0.66$ and the lateral shift is $\rho_i = 0.4$

Same maze configuration, different goal locations

**a** Trajectories in the maze    **b** Euclidean hypothesis    **c** Graph hypothesis

Figure S4: Trajectories in behavioral and latent spaces for example trajectories to three goals in maze configuration 1. (a) Trajectories in the physical maze. (b) Trajectories in latent space for neural firing generated according to the Euclidean hypothesis. Shaded colored regions are the union of all generated trajectories. (c) Trajectories in latent space for neural firing generated according to the graph hypothesis.

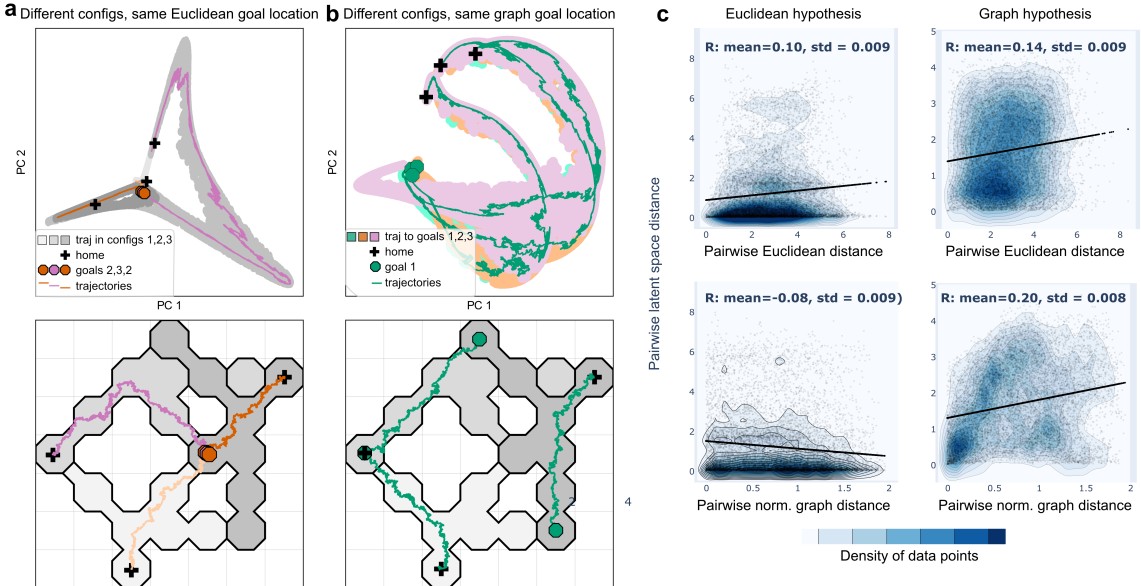

Figure S5: Comparative analysis in the spaces spanned by the first two principal components of PCA. **(a)** Under the Euclidean hypothesis, PCA partly captures the Euclidean structure of the data. The common Euclidean location in the maze apparatus also appears as one common location in the PC space. Maze configurations appear to be separated in the latent space. **(b)** Under the graph hypothesis, PCA fails to capture the underlying structure of the data. Latent states corresponding to trajectories towards different goals highly overlap. Trajectories leading to the same goal in different maze configurations appear as relatively far from each other in the latent space. **(c)** Comparison between pairwise PC space distances and pairwise Euclidean and normalized graph distances for 100 sets of $10^4$ pairs of points sampled from each dataset. Better correlation is achieved using the Euclidean (normalized graph) distance for the Euclidean (graph) dataset, however, the correlations are similar and fairly low for both metrics and we can no longer claim that the standard metric in the learned latent space reflects the choice of the neural representation.

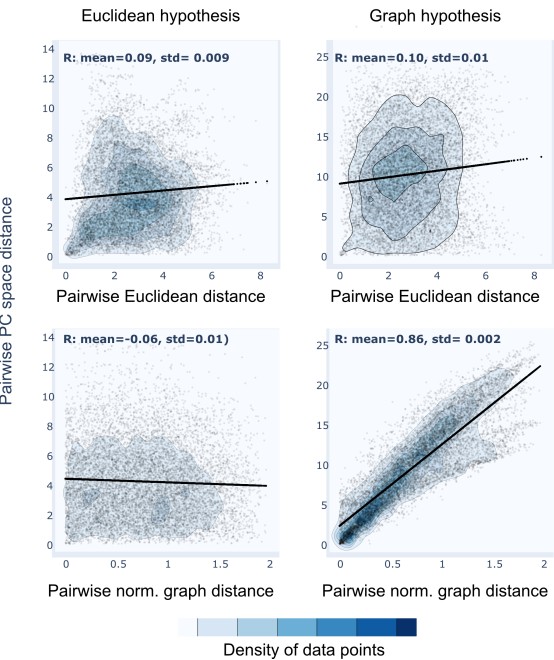

Figure S6: Impact of noise on the pairwise latent space distances from UMAP, plotted against pairwise Euclidean and normalized graph distances. Compared with the AE (Figure 2), a better correlation is still achieved using the normalized graph distance for the graph dataset, and UMAP might still be suitable to discriminate between the hypotheses. However, in the Euclidean dataset, the correlations are low for both metrics, so the learned latent space does not reflect the underlying structures of the neural representations.

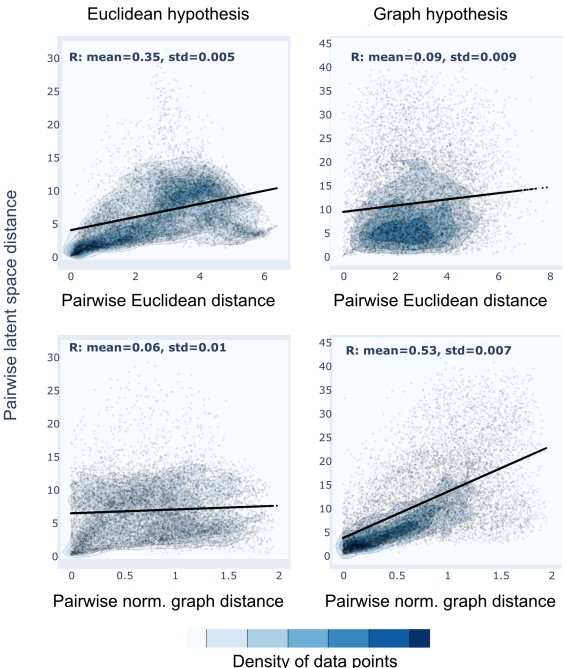

Figure S7: Impact of noise on the pairwise latent space distances from AE, plotted against the pairwise Euclidean and normalized graph distances. The standard metric in the learned latent space still reflects the underlying structure of the neural representation.

