# OpenReview forum: "Testing geometric representation hypotheses from simulated place cell recordings"
_NeurIPS.cc/2022/Workshop/NeurReps — NeurReps 2022 Poster_

### Official Review · Reviewer_kZHB · 2022-10-14
**Review on discovering topology in place cell recordings**

**Confidence:** 4
**Soundness:** 3
**Presentation:** 2
**Contribution:** 2
**Overall Rating:** 6

**Summary:**

In this extended abstract, the authors propose to test whether autoencoders (AEs) outperform PCA in differentiating between Euclidean and geometric representational hypotheses. They use the example of place cell recordings. It is well known that hippocampal place cells represent more than an animal’s position in Euclidean space, including task related activity. Here, the authors propose an approach to distinguish between “Euclidean” and “Graphical” representations, where graphical representations potentially could better support representing task variables. To test whether AEs or PCA better distinguished between these hypotheses, the authors simulated random runs through 3 different mazes, constructed by connecting tiles within a 7x7 grid. Note, the three mazes share topology, but not necessarily Euclidean space or tile distances between home and goals; they each have 1 home position and 3 arms to 3 goals.

Next, they ran a total of 270 simulated runs through the maze (15 trajectories x 3 goals x 3 mazes x 2 (Euclidean and graphical models)). In each simulated run, they “recorded” the activity of 100 neurons that were placed across the 7x7 grid. It should be noted that for all Euclidean runs, the 100 neurons were in the same position and for all “Graphical” runs, the 100 neurons were redistributed to lie within the maze and “preserve the same relative position in the graph”.

Finally, both AE and PCA can be used on the recorded data to determine latent spaces of the representation. Using latent spaces discovered by AEs and PCA, the authors compared the actual (in maze) and latent Euclidean and a normalized graphical distances between pairs of points sampled within each dataset (i.e., from each model run). They found significant correlations between maze distances and its latent representation in AE space, but not PCA space. From this result, they drew the conclusion that AEs are better at distinguishing between underlying representational hypotheses than PCA.


**Questions:**

Several points regarding the representational models require clarification. Overall, these questions all relate to me wanting more information about how the Euclidean and Graphical Representations are actually different. Perhaps, because of the same limitation the authors were not fully able to show all details of the model and for that reason I am unable to really evaluate how the models are different and in what ways those differences may or may not interesting or of interest to a broader audience.

The difference between topology vs. graphical model. I could be misunderstanding, but the topology of the maze, as implied by the authors, reflects the relative positions of task points within the maze: the home, choice point and goal points. Meanwhile, the graphical model is based on firing rates that depend on the Euclidean distance for the most part and or “tile distances”, neither of which are directly related to topology? The authors seem to be confusing / conflating these terms.

I am unsure that I understand exactly what the difference is between the FR responses in Euclidean and graphical models. Looking at the responses in Figure 1, I cannot visually see a difference between the place fields WITHIN the maze. It would help to see a side-by-side comparison of a neuron in the same Euclidean spot within the same maze, but with either a Euclidean or Graphical FR function.

To me, it is unclear from figure 1c which neurons from which runs are being plotting. It might help to show at least two plots (one per model – Euclidean and Graphical) of all 100 neurons from one configuration x trajectory for a clear side by side comparison.


**Limitations:**

The authors discussed the limitations of their work, namely that they would like to explore more models of both the latent space and representational space. Additionally, they mentioned that they should create a more realistic model of neural activity to be more complex than the simple Euclidean model. I agree with them and hope they get the opportunity to pursue these interesting ideas further.

**Recommended Decision:**

2: Borderline

**Relevance:**

3: Solid fit

**Strengths And Weaknesses:**

The idea that place cell activity can be better understood by task space topology rather than Euclidean geometry of space is interesting to me. In general, I find it appealing to try to think about task structures according to their topology rather than physical features (e.g., Euclidean distances or positions). Thus, looking at whether a maze should be represented as a topological object rather than Euclidean is interesting to me. Furthermore, it would be cool to see how this might be done by neurons. This question is not what is addressed in this paper – yet perhaps, if it were this paper would be more novel and exciting.

Currently, the authors are essentially showing that AEs are better than PCA at finding the difference between geometric (here Euclidean and Graph based) representations. However, even that conclusion is not particularly interesting as AEs have been compared to PCA. If this is of interest to the authors, they may with to compare AEs to other non-linear manifold learning techniques. Furthermore, I am concerned, unless I am missing something, that this comparison between AE and PCA is unfair, given that overall neural selectivity is not constant across their models and AE might be more sensitive to changes in sample size compared to PCA. Note in Figure 1, we can see that the Euclidean representation has a sparse code (fewer samples or lower SNR) by virtue of the neuron placement in their mazes compared to the density of neurons in their Graphical representations.

Also, the authors may wish to expand on the differences between geometric representations. If so, they would need better explain and expand the Euclidean and Graphical representations which were only explored for one set of parameters and neural placements. They should explain what happens when they expand out or contract the number of tiles in the graphical distance metric, because right now the place fields between the two models look practically the same. The only difference I can see in Figure 1 is that the graphical representation will not fire beyond the maze walls, however this will never matter given that simulations will not run beyond the maze walls. Otherwise, given the parameters chosen for both models, they seem the same, which begs the question on why the AE is finding different latent representations.

Finally, and I could be missing something here, but I cannot understand why the authors chose to change Euclidean space between the configurations. It seems like they could have avoided a lot of trouble if they just kept the space the same Euclidean space (7x7 grid of tiles) and changed the topology of some (but not all) home to goal paths by adding in walls between tiles. This way they could have kept all neurons (and gross selectivity of the population to the task) the same across all models and directly compared how changing either the Euclidean or Graphical distance between home-goal changed the representations.


**Submission Track:**

Extended Abstract (4 Page)

---

### Official Review · Reviewer_LDfH · 2022-10-16
**Outline of an interesting approach needing more validation.**

**Confidence:** 3
**Soundness:** 2
**Presentation:** 3
**Contribution:** 2
**Overall Rating:** 5

**Summary:**

The authors address the question of whether place cells represent physical or task space by simulating the place-cell responses of a rat moving in a maze task. In one set of simulations place cell firing rates were determined by the Euclidean distance of the animal's position to the center of each each cell's place field, while in another set of simulations graph distance was used. They trained an autoencoder with a three-dimensional latent state on the place cell responses and compared the latent representations of the simulated rat's positions in the maze with the rat's actual positions in the maze. They found that Euclidean distances between latent representations of the rat's position were better correlated with Euclidean distances in the maze when Euclidean distance determined place cell firing rates, and with maze distances measured using graph distance when graph distance determined firing rates. When PCA was used instead of the autoencoder the same effect was observed though much weaker. The authors conclude that their auto-encoding approach may be useful for evaluating distance metrics hypothesized to underly the firing rates of real place-cells in the brain.


**Questions:**

1. Have the authors performed any statistical tests to back up their claims that e.g. distances in latent space are better correlated with the distance metrics that determined place cell firing? The authors also claim that their approach outperforms PCA, based on the higher correlations of the autoencoder latents with the distance metrics that determined the firing rates. However, there are no error bars on these correlations, so the effect they report for the AE may also be there for PCA, and would be much easier to compute.
2. The authors are interested in distingushing the distance metrics that drive place cell firing. Have the authors tried to answer this question directly by using e.g. Bayesian model selection?

**Limitations:**

The authors do acknowledge that their model needs more validation on e.g. different task geometries and more realistic firing rates. However to better establish their approach they need to provide statistics backing up their claims, and to also show that it outperforms more direct alternative approaches to answering their question about the distance metrics that determine place cell firing, such as model selection.

**Recommended Decision:**

2: Borderline

**Relevance:**

3: Solid fit

**Strengths And Weaknesses:**

The main strength of the paper is the simplicity of the approach. It would be easy to train the autoencoder on real place cell data and to then compare the pairwise distances of the latent representations to various task-derived distances to help determine which task parameter best determines place cell firing. This could be a useful tool for neuroscientists determining what drives place-cell firing.The main weaknesses of the paper are insufficient validation of the approach to determine its robustness and dependence on the various architectural and task choices made, lack of statistics quantifying the main claims, and lack of a comparison to more direct alternatives such as Bayesian model selection for determining the distance metrics that best explain place-cell-firing.

**Submission Track:**

Extended Abstract (4 Page)

---

### Official Review · Reviewer_5C4X · 2022-10-19
**Autoencoders better capture the geometry of representations of place cell populations**

**Confidence:** 4
**Soundness:** 3
**Presentation:** 3
**Contribution:** 2
**Overall Rating:** 4

**Summary:**

This paper presents some preliminary findings to understand the geometry of representations of place cell populations. More specifically, using simulated data that encode for positions (Euclidean or Graph-Based) of a rat in a maze, the structure of the latent space is learned by an autoencoder, and also analyzed using PCA. It is shown that the underlying geometry (whether it is Euclidean or graph-based) can be captured using auto-encoders, but PCA can't do so. The data is generated using a Maze apparatus and includes physical Euclidean locations, or locations in graph space, and also includes cases when the underlying graph topology remains the same but the physical Euclidean positions change. The details for data generation are specified in Section 2.1. The autoencoder design is plain-vanilla, and the results are described in Figure 2. Overall I think the investigation is quite preliminary. The premise of the paper is quite interesting, although the use of autoencoder suggests that the non-linearity helps unroll the data manifold (assuming it is a manifold) in a way that PCA can not, which isn't quite surprising. It is not entirely clear if the autoencoder can indeed capture the geometry -- in the sense that pairwise distances in the latent space represent the situation in the input space. If other factors (path integration, temporal dynamics) are also taken into account, I suspect that certain symmetry conditions will also need to be investigated to constrain the latent space (based on constraints on the input space).



**Questions:**

I don't have any questions.

**Limitations:**

Yes.

**Recommended Decision:**

1: Reject

**Relevance:**

4: Highly relevant

**Strengths And Weaknesses:**

- Well motivated problem, interesting experimental results, and quite well written.
- The work is quite preliminary, as of now. Since the ML novelty is certainly limited, it would have been far more interesting to see what other information can be squeezed out (from a neuroscientific perspective) from this set up itself.

**Submission Track:**

Extended Abstract (4 Page)

---

### Decision · Program_Chairs · 2022-10-21

**Decision:**

Accept (Poster)

**Comment:**

The following extended abstract has received an average score of 5 (4,5,6) -- above the default cutoff for acceptance of extended abstracts -- but with a borderline-to-reject (1,2,2) recommended decision.  Thus, we review the assessment.

We note that the main critiques are regarding the early stage of the findings (i.e. limited statistics, synthetic data 5C4X, LDfH, kZHB), and the limited novelty as a machine learning approach (5C4X).   The primary strength of this submission is as a tool for empirical neuroscientists: this work provides a novel method for designing arbitrarily complex non-Euclidean experimental environments and modeling that structure in the resulting neural representations. Reviewer LDfH notes that the tool is both simple and useful in this regard.

A further understanding of the latent space structure is necessary to establish how well the proposed architecture captures the underlying Euclidean or graph hypothesis (noted by kZHB, LDfH) -- although the correlations suggest promising early-stage results in that direction. Reviewer LDfH additionally notes the the lack of consideration of alternative models.

Despite these drawbacks, we assess the paper to be highly relevant both to the topic of the workshop and to computational neuroscience researchers more generally. Moreover, we find the demonstrations on synthetic data convincing enough to warrant inclusion as an extended abstract in the workshop, as the EA track is intended for technically solid, relevant, early stage work. We recommend that the authors consider the constructive critiques of the reviewers in future work.